# Biodiversity loss reduces global terrestrial carbon storage

Sarah R. Weiskopf ®[1,2] ✉, Forest Isbell ®[3], Maria Isabel Arce-Plata[4], Moreno Di Marco ®[5], Mike Harfoot[6], Justin Johnson ®[7], Susannah B. Lerman[8], Brian W. Miller ®[9], Toni Lyn Morelli ®[2,10], Akira S. Mori ®[11], Ensheng Weng[12] & Simon Ferrier[13]

Natural ecosystems store large amounts of carbon globally, as organisms absorb carbon from the atmosphere to build large, long-lasting, or slow-decaying structures such as tree bark or root systems. An ecosystem's carbon sequestration potential is tightly linked to its biological diversity. Yet when considering future projections, many carbon sequestration models fail to account for the role biodiversity plays in carbon storage. Here, we assess the consequences of plant biodiversity loss for carbon storage under multiple climate and land-use change scenarios. We link a macroecological model projecting changes in vascular plant richness under different scenarios with empirical data on relationships between biodiversity and biomass. We find that biodiversity declines from climate and land use change could lead to a global loss of between *7.44-103.14* PgC (global sustainability scenario) and *10.87-145.95* PgC (fossil-fueled development scenario). This indicates a self-reinforcing feedback loop, where higher levels of climate change lead to greater biodiversity loss, which in turn leads to greater carbon emissions and ultimately more climate change. Conversely, biodiversity conservation and restoration can help achieve climate change mitigation goals.

Climate change and biodiversity loss have been increasingly recognized as related crises that are most effectively addressed together[1-5]. Hundreds of experimental studies have consistently found that within a place, more diverse assemblages, and in particular more diverse plant assemblages, have higher standing biomass production and carbon sequestration[6-9]. There are several possible mechanisms for this phenomenon. Species with different traits and resource requirements may be able to utilize more resources in an ecosystem through

reduced competition, increased facilitation, or both, which leads to overall more efficient resource use[10-12]. At the same time, more diverse assemblages are more likely to contain the most productive species, which can increase overall functioning[13,14]. Indeed, biodiversity loss can be one of the major drivers of productivity loss within ecosystems, on par with elevated carbon dioxide or effects of drought[15]. Thus, while climate change can affect biodiversity, biodiversity loss can also affect climate change by altering carbon sequestration and storage[4,16].

[1]U.S. Geological Survey National Climate Adaptation Science Center, Reston, VA, USA. [2]Department of Environmental Conservation, University of Massachusetts, Amherst, MA, USA. [3]Department of Ecology, Evolution and Behavior, University of Minnesota, Saint Paul, MN, USA. [4]Département de Sciences Biologiques, Université de Montréal, Montréal, QC H3T 1J4, Canada. [5]Department of Biology and Biotechnologies, Sapienza University of Rome, Rome, Italy. [6]Vizzuality, 123 Calle de Fuencarral, 28010 Madrid, Spain. [7]Department of Applied Economics, University of Minnesota, 1994 Buford Ave, Saint Paul, MN 55105, USA. [8]USDA Forest Service Northern Research Station, Amherst, MA, USA. [9]U.S. Geological Survey North Central Climate Adaptation Science Center, Boulder, CO, USA. [10]U.S. Geological Survey Northeast Climate Adaptation Science Center, Amherst, MA, USA. [11]Research Center for Advanced Science and Technology, the University of Tokyo, 4-6-1 Komaba, Meguro, Tokyo 153-8904, Japan. [12]Columbia University/NASA Goddard Institute for Space Studies, 2880 Broadway, New York, NY 10025, USA. [13]CSIRO Environment, Canberra, ACT 2601, Australia. ✉e-mail: sweiskopf@usgs.gov

Despite the contribution that biodiversity itself makes to carbon sequestration, high-level nature-based solution initiatives often focus on increasing the spatial extent of natural ecosystems, particularly forests, and not on their diversity or composition[17].

Similarly, ecosystem service models do not always account for the effects of biodiversity. Models projecting changes in biodiversity, ecosystem functioning, and ecosystem services typically operate independently and do not account for interactions or feedbacks[18,19]. For example, Earth System Models (ESMs) typically model terrestrial ecosystems using a small number of plant functional types and do not include biodiversity-productivity mechanisms[20,21]. Not accounting for biodiversity may lead to inaccurate projections of ecosystem function and ecosystem services. For instance, these estimates assume that remnant habitat patches will provide the same level of function even in the face of significant losses of species diversity[22]. Incorporating biodiversity-ecosystem function relationships could improve model accuracy, especially over long timescales as biodiversity effects become stronger over time[23]. For example, an Australian ecosystem modeling exercise found that including species turnover in marine ecosystem models led to very different outcomes for marine fisheries under different climate change scenarios[24].

Multiple pathways to integrate biodiversity and ecosystem function models have been proposed[25]. One approach that can be applied at the global scale is to connect biodiversity to ecosystem function and ecosystem service models using empirical, observational, or experimental biodiversity-ecosystem function data. Because of the extensive experimental data on the increase in biomass associated with increasing plant species richness[9,26], assessing how loss of plant diversity will affect carbon storage offers a feasible and useful case study to demonstrate the utility of this modeling approach. Moreover, assessments of plant species diversity and carbon storage are relevant for monitoring biodiversity and climate change mitigation goals. Early

analyses have been conducted that illustrate this method[4,22]. For example, Isbell et al.[22] linked regional estimates of species loss (using species-area relationships) with biodiversity-ecosystem function relationships derived from local-scale experiments. Yet, that work did not consider how climate change or land-use change might affect spatial patterns of species compositional turnover. Species turnover and regional species richness likely have important effects on functioning and stability[27–29]. We build on this previous analysis by accounting for compositional turnover while estimating regional diversity loss.

We use the Biogeographic Infrastructure for Large-scaled Biodiversity Indicators (BILBI) model to project changes in plant species richness[30], going beyond the species-area[22] or endemics-area[4] relationships considered in previous studies by also accounting for variation in the species composition of communities (beta diversity) at fine spatial scales. This allows BILBI to be used to assess species persistence/loss over the long term under different scenarios of land-use and climate change[30,31]. We link our species-loss estimates with empirical biodiversity-biomass stock relationships[9] to assess the biomass loss, and ultimately carbon storage loss, associated with loss of vascular plant diversity. Although carbon stocks are affected by many global change drivers (e.g., climate, land use), this approach allows us to estimate carbon loss driven specifically by biodiversity loss. Like previous analyses[22], we use data from local-scale biodiversity experiments to estimate biodiversity-biomass relationships at regional scales. The advantage of using the local experimental data is that by strictly controlling for species richness, composition, and other confounding factors, local experiments can disentangle the causal effects of species richness on biomass production. This assumes that (1) local loss of species diversity is similar to regional scale biodiversity loss, and (2) species loss occurring at the regional scale has consequences for ecosystem functioning. Although the first assumption may not always be met, there is considerable evidence for the second assumption[28]. BILBI produces estimates of plant species loss, based on the local-scale effect of land-use change on species persistence and the regional-scale effect of climate change on species composition. Therefore, we followed previous analyses[22] and used estimates of plant species loss at the ecoregion scale. Our study explores how projected future climate and land-use change scenarios will affect biodiversity loss. We estimate the additional long-term loss of carbon storage resulting indirectly from biodiversity loss that is expected in addition to the direct emissions from land-use change or other climate change impacts on carbon stocks (Fig. 1).

We used BILBI model projections of the proportion of vascular-plant species expected to persist under "global sustainability" and "fossil-fueled development" scenarios that were produced for a recent model intercomparison project[32]. The BILBI model uses generalized dissimilarity models fitted with more than 52 million records from over 254,000 plant species to map beta diversity at ~1 km² scale globally (see refs. 30,31 for complete model fitting information). Following Weiskopf et al.[25] (pathway A), we combined beta-diversity estimates with species-area relationships to assess plant species losses in each ecoregion globally, and then used empirical estimates of biodiversity-biomass stock relationships from O'Connor et al.[9] to assess proportional changes in plant biomass. Finally, we used projected terrestrial carbon stock maps (which do not include biodiversity losses) from the Coupled Model Intercomparison Project Phase 5 (CMIP5) IPSL-CM5A-MR model to estimate aboveground plant and soil carbon storage losses associated with projected biodiversity loss in each ecoregion[33].

## Results

Under the global sustainability scenario, the 818 ecoregions lost an average of 16.0% of plant species using a $z$-value (species-area relationship) of 0.25, ranging from −14.6% to 45.9% for individual ecoregions (Fig. 2; see Supplementary Fig. 2 for full range of $z$-values). This led to an average biomass loss of 4.4% using a $b$-value (the power

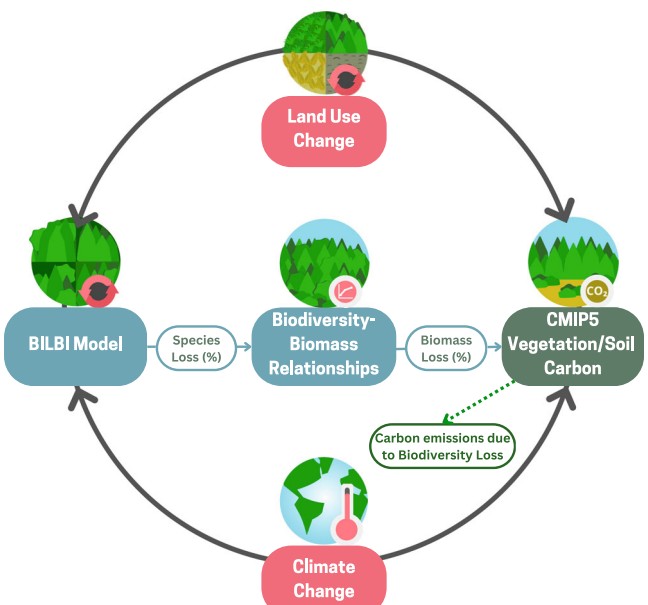

**Fig. 1 | Modeling framework for this analysis.** Coupled Model Intercomparison Project (CMIP) and Biogeographic Infrastructure for Large-scaled Biodiversity Indicators (BILBI) models were used to estimate the effects of land-use and climate change (red boxes) on vegetation/soil carbon and biodiversity, respectively. We used BILBI model output of proportional species loss and empirical biodiversity-biomass relationships to estimate the proportional biomass loss from biodiversity loss (blue boxes). We then applied this proportional biomass loss to soil and vegetation carbon estimates from CMIP5 (green box) using similar climate change and land-use change scenarios as the BILBI model to estimate carbon emissions due to biodiversity loss (green-dashed line).

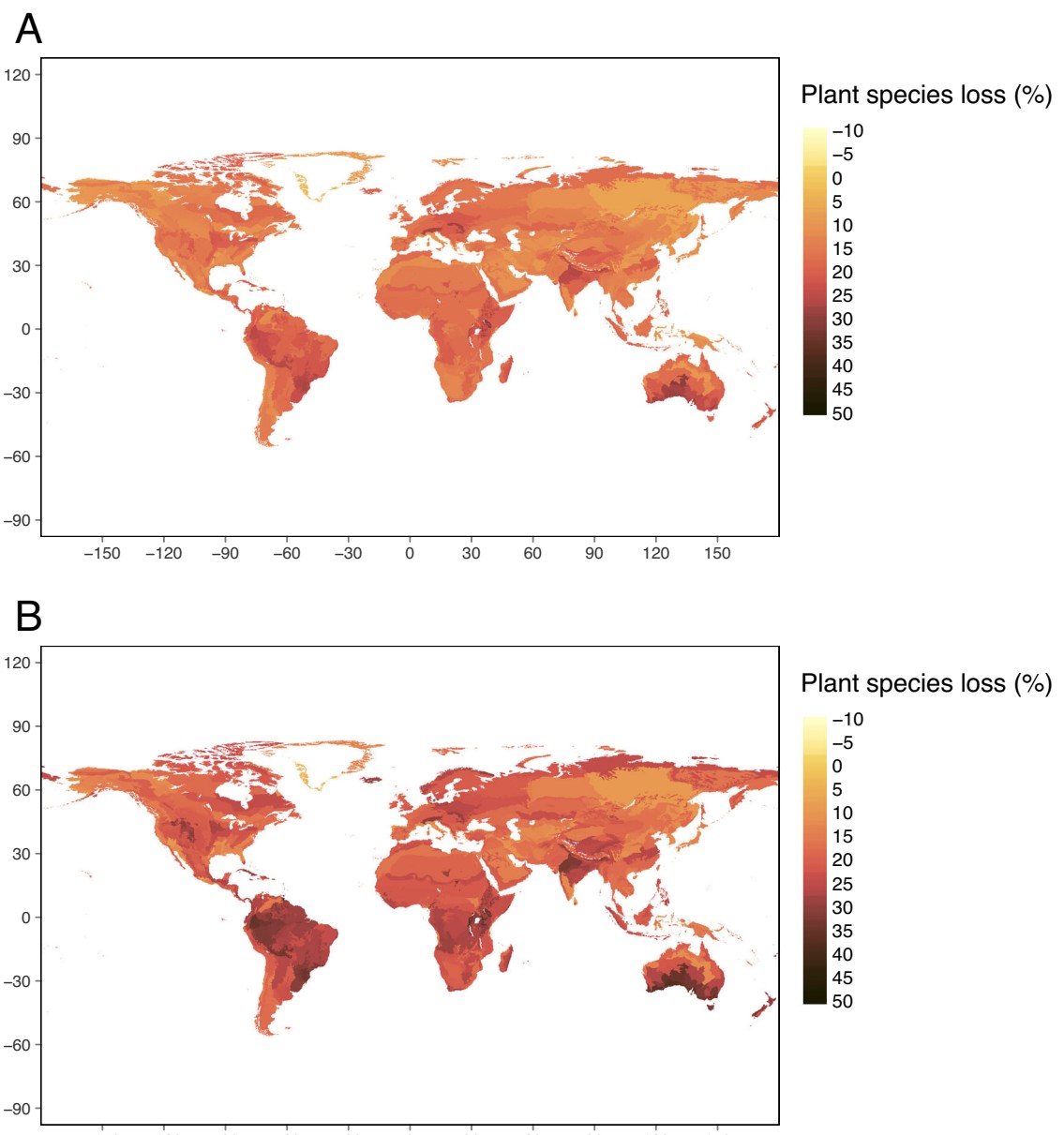

**Fig. 2 | Species loss by ecoregion.** Plant species loss by ecoregion projected by the Biogeographic Infrastructure for Large-scaled Biodiversity Indicators (BILBI) model under a global sustainability (SSP1/RCP2.6, **A**) and fossil-fueled development (SSP5/RCP8.5, **B**) scenario using a species-area relationship of $z = 0.25$. Darker areas indicate greater plant species loss. Species-loss estimates are what is expected over the long term, when ecosystems approach their new equilibrium states, based on climate and land-use changes projected for 2050.

relationship between a change in species richness and biomass) of 0.26, ranging from −3.6% to 14.8% (Supplementary Fig. 1, see Supplementary Fig. 4 for full range of $z$ and $b$ values). This biomass loss was from within remaining vegetation as a result of biodiversity loss, over and above any biomass loss resulting from the direct impact of land-use change under a given scenario. Biodiversity and biomass losses were higher under the fossil-fueled development scenario, with ecoregions losing an average of 20.8% of plant species (ranging from −36.9% to 46.2% across individual ecoregions; Fig. 2, see Supplementary Fig. 3 for full range of $z$-values), leading to an average biomass loss of 5.9% (ranging from −8.5% to 14.9%; Supplementary Fig. 1, see Supplementary Fig. 4 full range of $z$ and $b$ values). In both scenarios, plant species loss, and therefore proportional biomass loss driven by plant species loss were especially high in the tropics. Southern Australia,

eastern Europe, and some regions of South America also had high losses.

When we combined the biomass loss values with projected carbon storage maps, we found that overall vegetation carbon loss was greatest in the tropical regions of South America, central Africa, and Southeast Asia, which was driven by which regions store the greatest amount of vegetation carbon and by the level of biodiversity loss (Fig. 3). For example, biodiversity loss projections and vegetation carbon were both high in the Amazon, making this a hotspot of biodiversity-driven carbon loss. In contrast, biodiversity loss projections were also high in southern Australia, but because this region has lower vegetation carbon, it was not an area with high biodiversity-driven carbon loss.

When summed across all terrestrial ecoregions, biodiversity declines led to loss of 7.40–102.68 (29.55 using $b = 0.26$ and $z = 0.25$)

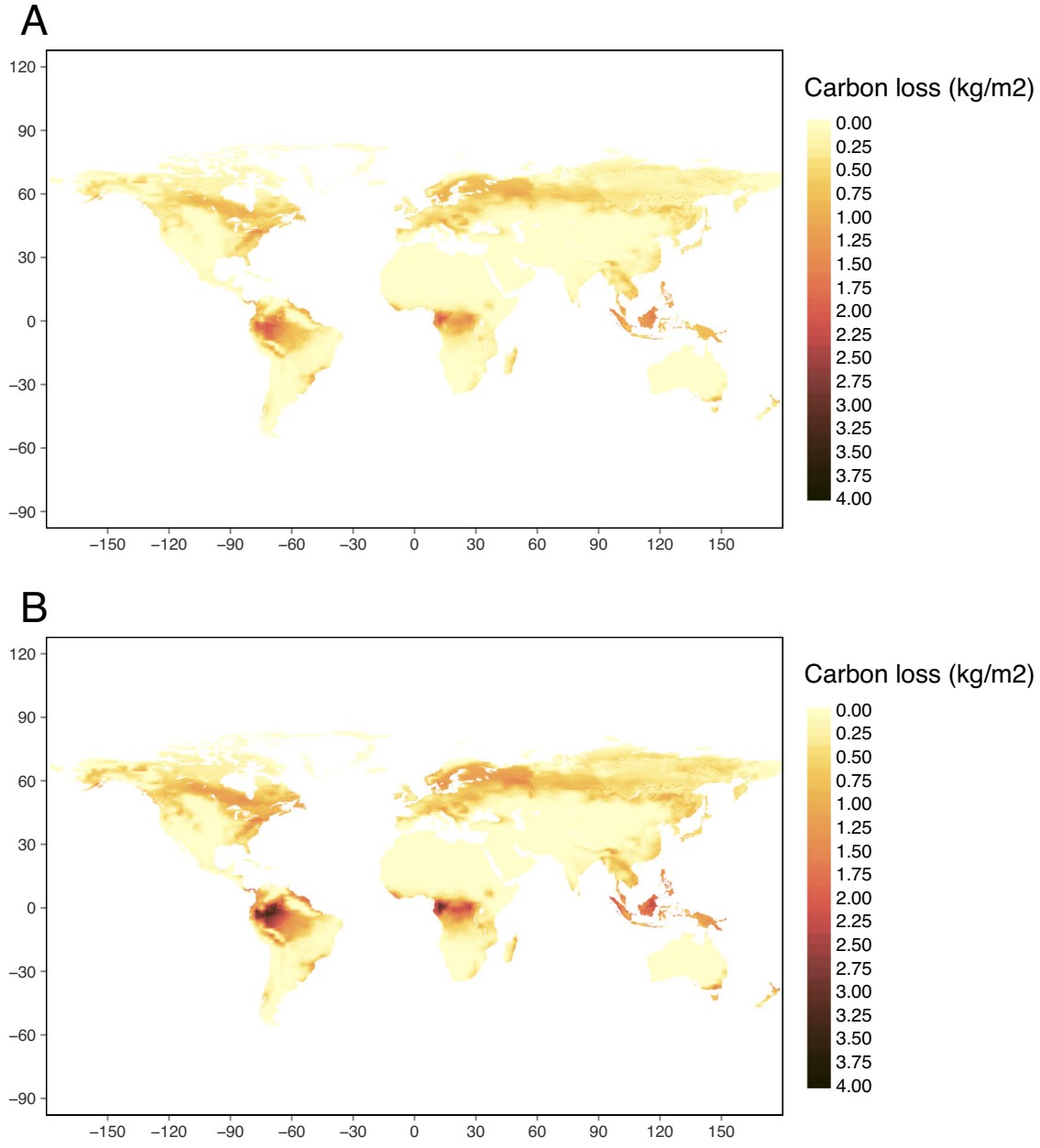

**Fig. 3 | Carbon loss by ecoregion.** Carbon loss (kg/m²) driven by long-term loss of plant biodiversity by ecoregion under a global sustainability (SSP1/RCP2.6, **A**) and fossil-fueled development (SSP5/RCP8.5, **B**) scenario using the mean biodiversity-ecosystem functioning slope $b = 0.26$ and a species-area relationship $z = 0.25$. Darker areas indicate greater carbon loss. This carbon loss is from within remaining habitat vegetation as a result of biodiversity loss, over and above any carbon loss resulting from the direct impact of land-use change (e.g., deforestation) under a given scenario. Carbon-loss estimates are what is expected over the long term, when ecosystems approach their new equilibrium states, based on climate and land-use changes projected for 2050.

PgC of vegetation carbon in the long term under global sustainability and 10.83–145.32 (42.89 using $b = 0.26$ and $z = 0.25$) PgC under fossil-fueled development. Again, this refers to losses within remaining habitat, above those resulting from direct impacts of land-use change on vegetation extent. The range of carbon-loss values was estimated from the full range of species-area relationships and biodiversity-biomass stock estimates, thus capturing a large range of uncertainty in the strengths of these relationships within and among sites (Supplementary Figs. 5 and 6). These carbon losses per ecoregion depended not only on how much plant diversity was lost from the ecoregion, but also the remaining area of the ecoregion, given that they are summed across all remaining habitat (Fig. 4). For example, under the global sustainability scenario, the overall loss of carbon was higher from the

ecoregions that have lost 10–20% of plant species diversity compared to ecoregions that lost >20% of their diversity because the former cover larger areas (Fig. 4).

Although our uncertainty range was high, projected carbon emissions from biodiversity loss have the potential to rival emissions expected from other sources such as land-use change or melting permafrost (Supplementary Table 1). Our models predicted long-term vegetation carbon emissions from long-term biodiversity loss (i.e., over the coming decades as the system moves toward a new equilibrium state) driven by climate and land-use change projections for 2050. These long-term biodiversity-driven carbon emissions were equivalent to about 12–169% of the total emissions expected from land-use change by 2100 for the global sustainability scenario, and

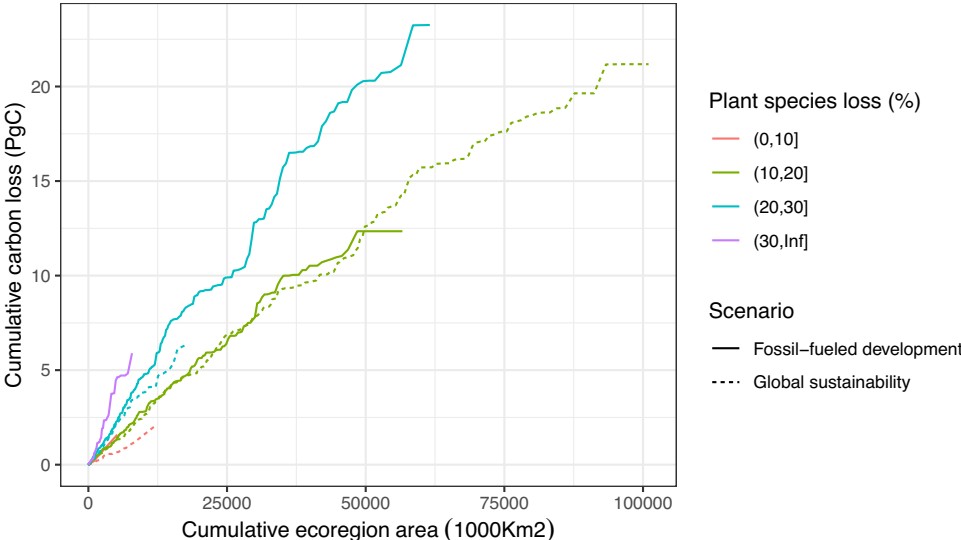

**Fig. 4 | Carbon loss by ecoregion area.** Cumulative carbon loss by cumulative ecoregion area (added from smallest to largest ecoregion size) grouped by proportion of plant species diversity lost in the ecoregion (depicted as different colors) for a global sustainability scenario (SSP1/RCP2.6, dashed line) and fossil-fueled development scenario (SSP5/RCP8.5, solid line). Because carbon losses are summed across all remaining habitat, places with moderate biodiversity loss, collectively, can contribute more to overall carbon loss than areas of high biodiversity loss. For the global sustainability scenario, carbon loss from ecoregions that lost more than 30% of plant species is <0.015 PgC and thus does not show up on the graph. Carbon loss estimates are what is expected over the long term, when ecosystems approach their new equilibrium states, based on climate and land-use changes projected for 2050. Source data are provided as a Source Data file.

about 20–271% for the fossil-fueled development scenario (Supplementary Table 1) using the full range of uncertainty from our analysis.

## Discussion

We used a macroecological model to predict changes in ecoregion-level plant species richness and linked it with empirical estimates of the plant biodiversity-biomass stock relationship based on experimental data. We found that biodiversity loss can reduce global carbon storage potential, and although our uncertainty range was large, it could lead to high loss of vegetation carbon. Substantially greater loss is projected under the more intense climate change and land-use change scenario, but even a sustainability scenario (compliant with the Paris target of 2 °C) carries high risks, similar to findings for mammals and wilderness areas[34,35]. This engenders a positive feedback wherein higher levels of climate change leads to greater biodiversity loss, which in turn leads to even greater carbon emissions.

Our study builds on a previous analysis that found that land-use change to date could result in the gradual loss of 2–21 PgC as plant species are lost from remaining habitats[22]. We found that projected emissions under future climate and land-use change have the potential to be much higher. In our analysis, we modeled aboveground carbon loss resulting from plant biodiversity loss. Soil carbon may also strongly depend on plant diversity[36–39]. Carbon loss could increase dramatically if the relationship is on a similar scale to aboveground biomass (18.71–259.72 PgC under the global sustainability scenario and 26.25–353.47 PgC under the fossil-fueled development scenario, Supplementary Figs. 7 and 8). A recent analysis using a different approach—linking species distribution models and other ecological modeling approaches with biodiversity-productivity relationships—found that mitigation activities that maintain tree diversity could avoid a 9–39% loss of productivity across terrestrial biomes[4]. Similarly, our estimated carbon loss from biodiversity loss was about 30% lower in the global sustainability scenario compared to the fossil-fueled development scenario. Our analyses used different biodiversity models and climate models, and Mori et al.[4] used estimates of local species loss rather than ecoregion losses as we did here. Together, these findings indicate that biodiversity loss can be a strong driver of carbon emissions.

Priority areas for biodiversity conservation and climate change mitigation could change by accounting for the role of biodiversity in promoting carbon storage. For example, Soto-Navarro et al.[2] identified few areas in central Africa that overlapped as being in the top 20% for both biodiversity and carbon importance. However, we found that carbon losses due to biodiversity loss were high in this area (Fig. 3), and therefore that biodiversity protection and restoration here could be highly valuable for climate change mitigation[40]. Several other factors that could contribute to this difference are that we used projected changes in biodiversity and carbon compared to current maps used by Soto-Navarro et al.[2], and also that we considered only plant diversity while Soto-Navarro et al.[2] looked at mammals, birds, and amphibians. Projected biodiversity loss and associated proportional biomass loss were higher in the Amazon and central Africa under the fossil-fueled development scenario compared to the global sustainability scenario. Interactions between biodiversity loss and ecosystem-level carbon storage led to consistently high losses of carbon in the tropics under both scenarios, specifically in the Amazon, central Africa, and Southeast Asia, and moderately high losses in boreal forests. In other words, because these places store large amounts of carbon, even smaller biodiversity losses under the global sustainability scenario can lead to significant overall loss of carbon. These places thus represent potential hotspots in terms of biodiversity and carbon storage loss.

Earth system models generally project increasing terrestrial carbon accumulation in high latitudes and decreasing accumulation in the tropics[41]. In our analysis, we found that northern latitudes may also experience carbon losses due to biodiversity loss. This could happen in part because our biodiversity model provides a more conservative estimate of potential species gains in these areas (discussed more below). Similar to Mori et al.[4], we found that total carbon loss from biodiversity loss was also greatest in the tropics (driven by the interaction between biodiversity loss and the location of high carbon stores), which may represent additional losses not captured in current models. Moreover, when proportional loss is considered, other areas such as southern Australia and the European Alps become hotspots of biodiversity and carbon loss (Supplementary Fig. 1). A recent intercomparison of ecosystem function models, including dynamic global

vegetation models, found similar patterns of carbon loss across South America and central Africa[42]. These models also found high losses in northern Africa, but northern Africa did not come out from our models as a hotspot of biodiversity-driven carbon loss. Interestingly, the model intercomparison found little difference in total ecosystem carbon between global sustainability and fossil-fueled development scenarios, likely due to $CO_2$ fertilization with higher levels of climate change[42]. Dynamic global vegetation models represent global plant diversity as a small set of plant functional types and simulate their distribution and biogeochemical cycles across the world under different climate and land-use change scenarios. Thus, these models are not accounting for how changes in species diversity within an area will affect biomass. Incorporating biodiversity-biomass relationships could be a useful way to improve such models in the future.

The IPCC estimates that the remaining carbon budgets—the amount of carbon that can be emitted by human activities while still limiting global warming to specified levels—is 140 PgC for limiting warming to 1.5 °C, and 310 PgC for limiting warming to 2 °C, although there is substantial uncertainty around these estimates[41]. While the uncertainty range for biodiversity-driven carbon loss is large, our high-end estimates for carbon loss from biodiversity loss constitute a large proportion of these limits (102.68 PgC under the global sustainability scenario and 145.32 PgC under the fossil-fueled development scenario). Not considering biodiversity loss in emissions scenarios could lead to severe overestimates of terrestrial carbon stocks and remaining carbon budgets.

Overall, our analysis points to the important role that maintaining and/or enhancing the diversity of plant species within areas of natural vegetation, in addition to increasing the extent of these areas, can play in addressing climate change. Alongside increasing the global extent of conservation areas (to prevent rapid carbon loss from ecosystem degradation), increasing plant species diversity in degraded ecosystems can increase carbon storage potential[3]. However, existing international initiatives like the Bonn Challenge and the Paris Agreement focus on forest extent rather than forest quality and composition for protection, afforestation, and reforestation[17,43]. Further, initiatives that include biodiversity goals do not always provide clear definitions of what constitutes a biodiverse restoration[44]. This can lead to planting monocultures with non-native species, which could be detrimental to biodiversity and carbon storage over the long-term[17]. Higher biodiversity, with the right species in the right places[45], could even help reduce the impacts of climate change on biodiversity, and therefore indirectly help maintain carbon storage potential of ecosystems[46].

Although informative, there are a number of uncertainties and limitations in our analysis that should be refined in future assessments. First, our empirical relationship between biodiversity and biomass stock comes from a meta-analysis of hundreds of experiments conducted at the local scale[9]. Experiments can disentangle the causal effects of species richness on biomass production. However, experiments generally take place over small spatial and temporal scales and may miss important processes like dispersal, evolution, and natural patterns of species assembly and loss[47]. This makes results more difficult to generalize to natural ecosystems, and additional work is needed to do so. See the Pathway A assumptions and challenges section in[25] for more discussion on this limitation. Moreover, the local scale of experimental data does not directly match the ecoregion scale of the BILBI model analysis. As discussed above, this assumes that (1) local loss of species diversity is similar to regional scale biodiversity loss, and (2) species loss occurring at the regional scale has consequences for ecosystem functioning of a similar magnitude to those for species loss at a local scale. There are several theoretical reasons why we expect biodiversity-ecosystem functioning relationships observed at a local level to be equally strong, and perhaps even stronger, across larger spatial extents. Larger spatial and temporal extents will encompass a greater range of environmental conditions.

This provides greater opportunity for niche partitioning, and thus positive biodiversity-ecosystem functioning relationships[28,48]. Additionally, whole landscapes require more species to maintain ecosystem functioning than do individual locations, with more diversity needed at broader spatial and temporal scales[49]. We presented estimated carbon losses over a large range of potential biodiversity-ecosystem functioning relationship values to capture some of the uncertainties introduced by these assumptions.

Second, the BILBI model assumes that if changes result in non-analog climatic conditions, species will not persist (and thus does not allow for adaptation or tolerance of conditions not experienced at present) and it also does not consider the possibility of increasing species richness in some ecoregions if species are able to exploit new habitat conditions as the climate becomes more suitable. Thus, the model presents a somewhat pessimistic estimate of biodiversity loss from climate change, a common issue with many species distribution model approaches[50–52]. However, native species assemblages have greater complementarity than exotic species assemblages due to longer histories of interactions. Thus, increasing species richness by adding species not previously present in the ecosystem may have a relatively small effect on productivity and may even decrease productivity or decrease the effects of biodiversity on productivity[53–55].

Third, it is important to correctly interpret the findings from our analysis. The BILBI model uses the species-area relationship to assess plant species persistence, meaning that it projects plant species losses expected in the long term due to habitat conditions in a given year (e.g., poor conditions in 2050 might generate losses beyond 2050). Because the BILBI model does not predict exactly how long it will take for species to disappear once environmental conditions have changed, we do not have an exact date for the projected changes in plant persistence. Therefore, our carbon storage loss estimates are also what is expected over the long term, when ecosystems approach their new equilibrium states, based on climate and land-use changes projected for 2050, whereas land-use and permafrost emissions were estimated from climate and land-use changes from present conditions up to 2100. Although long term is not easily defined, the way that species loss scales with area becomes larger over longer time frames[56]. That is, some species will disappear right away when they lose all suitable habitat, whereas others may disappear over time as remaining habitats are not able to sustain viable populations. Similarly, the effects of biodiversity grow stronger (and less saturating) over time[23]. Thus, estimates produced using smaller species-area ($z$-values) and biodiversity-biomass production estimates are more likely over shorter timescales, while larger losses become increasingly likely as more time elapses. By using a range of species-area relationship values, we attempted to capture the range of future biodiversity-loss-driven emissions that might be seen over different time scales.

Finally, our estimates of total carbon loss are based on projected carbon maps from a single general circulation model from CMIP5 (IPSL-CM5A-MR). Our goal was to compare scenarios with each other and provide a range of reasonable carbon loss estimates rather than absolute losses. Scenarios (including emissions and land-use) are a major source of uncertainty compared to global climate models when modeling persistence probability, and the selection of biodiversity modeling approach is also a major source of uncertainty[57,58]. We used projected carbon maps from the IPSL-CM5A-MR model to be consistent with our biodiversity model input parameters, but terrestrial carbon uptake estimates vary across CMIP5 and CMIP6 models[59]. Among CMIP5 models assessed, IPSL-CM5A-MR correctly reproduced the global land sink in comparison with historical data, but was not the best performing model for the cVeg variable that we used in this analysis[60]. Recent analysis found that IPSL-CM5A-MR produced estimates of near-present plant carbon within the range of observation-based estimates in the non-circumpolar region, but overestimated the

circumpolar regions[21]. Thus, our carbon loss estimates from biodiversity loss may also be overestimated in these regions.

Biological carbon sequestration and biodiversity are tightly linked (Fig. 5). Biodiversity-mediated carbon loss has the potential to rival emissions from other sources, so achieving Sustainable Development Goal 15 (Life on Land) can contribute to achieving Goal 13 (Climate Action)[61]. While meeting the Paris Agreement would prevent a large amount of carbon loss compared to a fossil-fueled economic development strategy, this scenario is still associated with potentially high carbon loss via biodiversity loss. Therefore, additional mitigation measures may be needed to meet Paris Agreement expectations even if current emission reduction targets are met. Improving our understanding of how biodiversity will adapt to climate change will be key to improving climate impact predictions. Carbon sequestration by naturally functioning ecosystems is an important element to offset the residual emissions that would occur even with maximum effort toward carbon neutrality.

Addressing climate change and biodiversity loss together will more effectively address these crises. Although policymakers are starting to think about climate change mitigation initiatives that have co-benefits for biodiversity, the role of biodiversity itself in promoting carbon storage is often overlooked, with much focus simply on biomass or ecosystem extent. On one hand, this may mean that the scientific community is underestimating future carbon emissions by not accounting for biodiversity-driven carbon losses, thus increasing the urgency for mitigating climate and land-use impacts. On the other hand, this highlights the important role that ecosystem restoration, focusing on the composition of these ecosystems, can play in climate change mitigation. In other words, there is potential to link the restoration target (T2) of the Convention on Biological Diversity (CBD) Kunming-Montreal Global Biodiversity Framework with that for climate-change mitigation (T8) and enhancing nature's contributions to people (T11), emphasizing a need to reconsider the functional value of biodiversity rather than focusing only on area-based measures for conservation (e.g., so-called 30 by 30; T3)[62]. At a national and local level, this could mean that a focus on maintaining and restoring diverse ecosystems can increase the return-on-investment for carbon storage over the same land area. This may be particularly important for those ecoregions that are projected to have high levels of biodiversity-driven carbon loss.

Our understanding of how biodiversity underpins ecosystem functions and services such as carbon storage has been increasing, but incorporating this knowledge into global projections and conservation policy lags behind[25,28,63-65]. Our modeling effort provides an important example of how we can effectively link biodiversity, ecosystem functions, and ecosystem services models. As our understanding of biodiversity and ecosystem function relationships improves, our analysis can be updated to reduce the uncertainty in the estimates. Building on and improving the modeling approach used in this study, including by filling the gaps identified in Table 1, can benefit ESM development and also help identify areas for conservation and restoration and thereby contribute to ongoing processes such as national biodiversity strategy and action plans under the CBD, nationally determined contributions for emissions reduction under the Paris Agreement, and payment for ecosystem services programs.

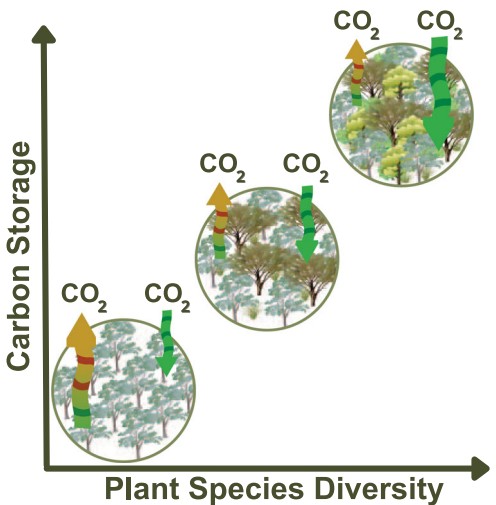

**Fig. 5 | Relationship between plant diversity carbon storage.** Conceptual graphic representing the role biodiversity plays in biological carbon sequestration. Increasing plant species diversity increases biomass stock. This is depicted as an increasing ratio between carbon sequestration (green arrows) and carbon emissions (yellow arrows). Illustrations from the Integration and Application Network, with no changes made. Images include: "Acer pensylvanicum", originally published by Joanna Woerner. Integration and Application Network (2010); released under a Creative Commons Attribution-ShareAlike 4.0 International (CC BY-SA 4.0, Acer pensylvanicum (Striped Maple) | Media Library | Integration and Application Network (umces.edu)). "Eucalyptus spp.", originally published by Lana Heydon. QLD Department of Environment and Resource Management (2008); released under a Creative Commons Attribution-ShareAlike 4.0 International (CC BY-SA 4.0; Eucalyptus spp. (Eucalypt) 1 | Media Library | Integration and Application Network (umces.edu)). Acer pensylvanicum (Striped Maple) | Media Library | Integration and Application Network (umces.edu)). "Acacia spp.", originally published by Kim Kraeer and Lucy Van Essen-Fishman. Integration and Application Network (2008); released under a Creative Commons Attribution-ShareAlike 4.0 International (CC BY-SA 4.0; Acacia spp. (Acacia) | Media Library | Integration and Application Network (umces.edu)). "Process; primary production", originally published by Tracey Saxby. Integration and Application Network (2003); released under a Creative Commons Attribution-ShareAlike 4.0 International (CC BY-SA 4.0; Process; primary production | Media Library | Integration and Application Network (umces.edu)). "Process; organic carbon release", originally published by Tracey Saxby. Integration and Application Network (2003); released under a Creative Commons Attribution-ShareAlike 4.0 International (CC BY-SA 4.0; Process; organic carbon release | Media Library | Integration and Application Network (umces.edu)).

## Table 1 | Knowledge gaps and future research directions

| Model component | Future research directions |
|---|---|
| Biodiversity model | • Incorporate species adaptation into future projections of persistence<br>• Project changes in local scale plant species richness that account for future climate and land-use changes to better match the scale of biodiversity-biomass production relationships |
| Biodiversity-biomass production relationship | • Collect more data on biodiversity-biomass production relationships in natural ecosystems to:<br>  ◦ better understand how the relationship varies across ecosystems to narrow the uncertainty range used in this analysis<br>  ◦ understand how processes operating over larger scales, such as dispersal, affect the relationship<br>  ◦ understand the effects of changing species composition in addition to changing species richness<br>• Assess the biodiversity-biomass relationship for soil<br>• Assess how plant functional traits, and thus biodiversity-biomass production relationships, will change under future climates |
| Carbon estimates | • Improve understanding of how productivity and carbon storage are affected by changing climates |

## Methods

### Step 1—Use BILBI model to estimate proportion of plant species expected to persist in each ecoregion under different climate and land-use scenarios

To assess how land-use change and climate change affect biodiversity, the BILBI model uses land-use data and projections to create a map of habitat condition, which is expressed in units of the proportion of native species expected to remain in each grid-cell, given the land-use type of that cell (Table S1 in ref. [31]). The model is also able to project climate-driven change in beta-diversity patterns, expressed in terms of the predicted dissimilarity (or conversely similarity) in species composition between any specified pair of grid-cells over both space and time. These projections are coupled with a modified form of species-area analysis to estimate the proportion of species expected to persist (i.e., avoid extinction) under a given scenario of land-use and climate change, within any given region. The use of a species-area relationship through generalized dissimilarity modeling (GDM) approach has been shown to be able to predict trends in biodiversity[31,66–70]. GDMs perform well as predictors of species composition[68,70]. The BILBI model relies on how species richness responds to land-use change locally, and this was tested (fit and validation) as part of the PREDICTS project[66]. Although more work is needed to understand the speed of extinctions following land-use change, species-area-based approaches have been found to be fairly good predictors of the number of threatened or extinct species[67,71]. See Supplementary Methods and refs. [30,31] for full model and model validation details, but briefly, this is achieved by:

(1) Calculating the total area of similar ecological environments relative to a given cell, by summing the predicted compositional similarity with all other cells under the present climate, and hypothetically assuming the habitat of all cells is in perfect condition.

(2) Calculating the potential area of similar ecological environments under a given future scenario, accounting for both the projected change in climate and the expected condition of habitat under that scenario.

(3) Expressing the effective area of habitat, across similar ecological environments, expected under a given scenario (from step 2 above), as a proportion of the total area of similar environments prior to climate and land-use change (from step 1 above, data available at [72]), and then using the species-area relationship to translate this proportion into the predicted proportion of species expected to persist over the long term. A species-area exponent of $z = 0.25$ was used in these calculations, as widely employed in other studies predicting the proportion of species expected to persist in fragmented habitats. However, intact habitats also experience species relaxation (i.e., long-term loss of species as the community approaches equilibrium species richness[73]), commonly estimated at $z = 0.15$. To estimate the additional loss of species due to climate and land-use change, we subtracted these two estimates of z to obtain a lower bound of $z = 0.1$[22,74]. We used a range of z values between 0.1 and 0.65, similar to Isbell et al.[22], to capture some of the uncertainty around the magnitude of species extinction debts.

Our scenario analysis followed the protocols laid out in Kim et al.[32]. We used two scenarios: SSP1/RCP 2.6 ("global sustainability"), a low land-use change and low climate change scenario which is compliant with the Paris target of keeping global warming to below 2 °C by the end of the century compared to pre-industrial times, and SSP5/RCP8.5 ("fossil-fueled development"), a high climate change and intermediate land-use change scenario[75,76]. Note that the global sustainability scenario still entails a significant amount of land-use change due to bioenergy production and increased food demand[77]. We chose these scenarios to represent the extreme low- and high-end outcomes to provide a full range of uncertainty estimates. We used land use data

from the 0.25° Land Use Harmonization dataset version 2 (LUH2)[78] and climate data from the 1 km WorldClim dataset[79]. For current conditions, we used the LUH2 data for 2015 and the WorldClim data for 1960–1990, and for the future conditions, we used LUH2 data for 2050 and WorldClim data for 2040–2060[31].

To obtain estimates of the proportion of species expected to persist at the ecoregion level ($p_{region}$), we used a weighted geometric mean of all cells in the ecoregion. The weight applied to each cell is inversely proportional to the total effective area covered by cells with a similar environment to the cell of interest. This means that cells within less extensive environments have a higher weight, since these areas are likely to support more unique species and thus are expected to contribute more to regional species persistence.

### Step 2: Use empirical relationships to link changes in species richness to changes in biomass

We use empirical biodiversity-biomass relationships from a recent meta-analysis based on 374 experiments (>500 entries from primary producers, dominated by terrestrial plant studies). They found general support for using a power function to describe how changes in species richness lead to changes in biomass for primary producers as follows[9]:

$$\text{Biomass} = a^{*}(\text{richness})^{b} \tag{1}$$

where $a$ is a constant representing the average biomass of a monoculture for the ecosystem, and $b$ describes the power relationship between a change in richness and biomass. As species richness increases, the biomass of the system will increase compared to the monoculture baseline, but the amount of increase per species decelerates as more species are added. This equation can be converted to proportion of remaining biomass ($p_{biomass}$) based on proportional change in species richness per ecoregion as follows:

$$p_{biomass} = (p_{region})^{b} \tag{2}$$

We apply this transformation to the BILBI model output to assess the proportion of remaining biomass from the proportion of remaining plant species richness, using the mean $b = 0.26$, as well as the 95% CI to provide uncertainty estimates around our results. O'Connor et al.[9] found that for primary producers, $b = 0.26$ (with a 95% CI of 0.16–0.37) was valid for most assemblages and was robust to differences in experimental design and the range of species richness levels considered. While they did not find an effect of study duration on b values, previous studies have found that biodiversity-productivity relationships grow stronger over time[23]. Although $b$ values can vary spatially[4,80], there is still uncertainty in how biodiversity-ecosystem functioning relationships differ across space and in how factors like climate, environmental conditions, and species trait compositions might systematically affect the observed relationship[9]. If the places where habitat destruction is highest are also the places that tend to have the highest or lowest biodiversity-ecosystem functioning relationships, then using a narrower range of spatially explicit values could systematically over or underestimate the carbon storage loss associated with this biodiversity loss. To address this concern, we estimated productivity losses associated with the full confidence interval range from O'Connor et al.[9]. Thus, rather than considering a single slope for the biodiversity-ecosystem functioning relationship, we consider a range of relationships that reflects variation in both composition and site-to-site differences found among previous biodiversity experiments. For example, the range of relationship values is wider than spatially explicit values estimated from in-situ forest re-measurement data globally (range = 0.198–0.299, mean = 0.26)[80]. Therefore, our range of $b$ values provides a conservative range of estimates of productivity loss associated with biodiversity loss. In addition, although estimates come from historical data, we do not

currently have estimates of how relationships may change in the future. Experimental evidence suggests, however, that positive biodiversity-productivity relationships are robust to droughts and changes in nutrient availability[81].

**Step 3: Estimate total changes in carbon storage and compare to other global change drivers**

The previous step provided spatially explicit estimates of proportional change in biomass associated with loss of biodiversity for each scenario. To convert biomass change to carbon storage change, we multiplied the gridded estimates of proportional change in biomass from the BILBI model by a global map of terrestrial carbon stock from CMIP5. The CMIP model estimates changes in carbon storage from climate change and land use change, but does not account for changes in species richness. By using these model projections as our baseline carbon estimate, we can assess the effects of biodiversity loss on carbon storage that are expected on top of the direct changes from climate change and land use change that have already been incorporated in initial carbon storage projections.

We used terrestrial carbon storage maps that considered only vegetation carbon, as well as maps considering vegetation and soil carbon. We calculated the average cVeg and cSoil value over a 12-month period in 2050 (the end year for the BILBI model output). Model inputs were not exactly the same, making it difficult to be consistent with the climate and land use input data used to estimate biodiversity loss and carbon storage. We downloaded the total carbon in vegetation (cVeg) and total carbon in soil (cSoil) layer from the CMIP5 IPSL-CM5A-MR model[33]. The biodiversity and ecosystem services models using harmonized scenarios (BES-SIM) used climate data from either the lower resolution IPSL-CM5A-LR or 1 km WorldClim data downscaled from the IPSL-CM5A-LR depending on biodiversity model requirements[32]. We chose to use the mid-resolution $1.25° \times 2.5°$ CMIP5 IPSL-CM5A-MR model to obtain higher resolution carbon maps. We obtained cVeg and cSoil for both of our scenarios—global sustainability (SSP1/RCP 2.6) and fossil-fueled development (SSP5/RCP8.5) from the Earth System Grid Federation (ESGF; https://esgf-node.llnl.gov/search/cmip5/). The BILBI model used land use data from the Land Use Harmonization dataset version 2[78], while the CMIP5 IPSL-CM5A-MR model used land use data from the Land Use Harmonization dataset version 1[82]. Version 2 provides higher resolution data over a longer time frame with more detailed land-use categories[78]. Although the input data are slightly different, we do not believe this invalidates the approach, as we are not comparing carbon storage changes between the models but instead using the CMIP5 IPSL-CM5A-MR model to estimate potential carbon storage under the global sustainability and fossil-fueled development scenarios, which we then use to estimate the possible magnitude of carbon storage loss driven by biodiversity loss under the same scenarios.

Soil type, climate, and land use are important drivers of soil carbon. Plant diversity can also increase soil carbon, and these relationships grow stronger over time[37-39]. If soil carbon depends strongly on plant diversity, then it is important to consider the possible magnitude of plant loss on soil carbon, even if the strength of these relationships is not fully established. We assessed how large soil carbon losses could be if plant biodiversity-soil carbon relationships are on a similar magnitude to aboveground biomass. We thus consider the additional impact that plant loss may have on soil carbon on top of the local environmental conditions considered in the CMIP models. We excluded soil types that are more likely to be impacted by drying and warming than by changes in plant diversity, including wetland (Gleysols), peatland (Histosols), and permafrost (Cryosols) soils[22]. We note that these soil types represent major global carbon stores. If biodiversity does in fact drive carbon loss in these ecosystems, we could be missing additional losses. Specifically, we resampled the 250 m predicted World Reference Base 2006 subgroup soil classification (ISRIC,

https://data.isric.org/geonetwork/srv/eng/catalog.search#/metadata/5c301e97-9662-4f77-aa2d-48facd3c9e14[83]); to the same resolution as the cSoil raster layer using the nearest neighbor method in the R software program *terra* package[84], and then masked out these soil types from the cSoil raster.

We did not account for potential changes in litter carbon. Increasing biodiversity increases the rate of litter decomposition (i.e., less litter mass storage), which could add to increasing decomposition from warming, and thus we would expect biodiversity loss to increase litter carbon storage. While the strength of the biodiversity-carbon relationships for soil and litter are not fully established, the effects on litter carbon are likely weaker than those on plant biomass or soil carbon[6,85,86]. For example, decomposition was 34.7% higher in mixed species forests compared to monocultures, while soil carbon storage was 178% higher in mixed grasslands than in monocultures[39,86]. Moreover, the estimated effects of diversity on plant biomass and soil carbon were driven by short-term studies, and these relationships grow stronger over time in long-term experiments[23,36,39].

To obtain cVeg and cSoil values on the same scale as the biodiversity data, we resampled by ecoregion using bilinear interpolation. Then, we multiplied our raster layers (proportion of remaining plant biomass and 2050 carbon maps), to obtain changes in carbon storage in 2050 in kg/m² ($\Delta C$), such that:

$$\Delta C = cVeg - cVeg^*(p_{biomass}) \text{ for vegetation carbon only, or}$$
$$\Delta C = (cVeg + cSoil) - (cVeg + cSoil)^*(p_{biomass}) \text{ for vegetation carbon and soil carbon.}$$

$$(3)$$

To convert this to total $C$ storage in PgC ($C_{total}$), we used the cellSize function in the *terra* package[84] to calculate the total area in m² of each ecoregion ($A$). We then multiplied this by the carbon storage layer to obtain total carbon storage lost per ecoregion, which we summed to obtain global $C$ storage loss values:

$$C_{total} = 1.0E^{-12} \sum_{k=1}^{n} \Delta C_k {}^* A_k \quad (4)$$

where $k$ = a given ecoregion and $n$ = total number of ecoregions.

We conducted all analyses in R version 4.1.1[87], and produced all graphics using either the *tmap* or *ggplot2* packages[88,89] (Supplementary Software 1–6).

## Data availability

BILBI model data are available at https://doi.org/10.6084/m9.figshare.25188650. CMIP data are available from the Earth System Grid Federation (ESGF; https://esgf-node.llnl.gov/search/cmip5/). World Reference Base 2006 subgroup soil classification data are available from ISRIC, https://data.isric.org/geonetwork/srv/eng/catalog.search#/metadata/5c301e97-9662-4f77-aa2d-48facd3c9e14[83]. Raster data for the maps generated in this study have been deposited in ScienceBase at https://doi.org/10.5066/P13WUFMU[90]. Source data are provided with this paper.

## Code availability

R scripts are included as Supplementary files.

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

## Acknowledgements

This work was supported by the National Socio-Environmental Synthesis Center under funding received from the National Science Foundation (grant no. DBI-1639145). A portion of this research was supported by the US Geological Survey National and North Central Climate Adaptation Science Centers. MDM received support from the European Union–NextGenerationEU as part of the National Biodiversity Future Center, Italian National Recovery and Resilience Plan (NRRP) Mission 4 Component 2 Investment 1.4 (CUP: B83C22002950007). We thank the Beth Fulton, the Diversity and Eco-Function working group, and the Morelli lab group for their feedback. We thank Alexey Shiklomanov for R coding assistance. Any use of trade, firm, or product names is for descriptive purposes only and does not imply endorsement by the US Government.

## Author contributions

S.R.W., F.I., and S.F. conceived the project. S.R.W., F.I., S.F., M.I.A.P., M.D.M., M.H., J.J., A.S.M., and E.W. contributed to the design of the analysis. S.R.W. led the analyses and wrote the initial paper draft. All authors, including B.W.M., S.B.L., and T.L.M., contributed substantively to revisions.

## Competing interests

The authors declare no competing interests.
