## [Peer Review File · Nature Communications]

Biodiversity loss reduces global terrestrial carbon storageREVIEWER COMMENTS

Reviewer #1 (Remarks to the Author):

The subject of the manuscript entitled “Biodiversity loss reduces global terrestrial carbon storage” submitted to Nature Communications journal is very attractive and falls into the most current research issues. This study can be an inspiration for further, more detailed research.

Below I leave some suggestions/comments that may allow the authors to improve this study:

- The authors need to better identify the research gaps and clarify what are its innovative contributions to science.

- The authors should mention the validation process of this study.

- It is missing to indicate in Figures 2, 3, and 4 the period to which the respective losses refer.

- The maps in Figures 1, 2, and 3 lack a graphic scale

- In the discussion section the authors should improve the following aspects:

(i) justify the results achieved;

(ii) explain why these results are similar or different from previous findings; and

(iii) highlight open questions and future research directions/challenges.

- The conclusions section should be expanded. More specifically, the authors should emphasize the contribution and implication of the study to science and recognize how this study can be useful for better land use management at the national/ regional level (and how it can be applied).

Reviewer #2 (Remarks to the Author):

This paper addresses a very important topic, namely the effect of biodiversity loss on ecosystem functions and services, which is the emerging field that highlight the impacts of losing biodiversity to mankind. The approach is novel as the authors link a state of art model for biodiversity with new insights of the role of biodiversity and standard models and data for carbon storage at a global level. This is not only challenging but also highly ambitious. The approach first estimates changes of plant species

richness related to land use change and climate change, using the BILBI model, then connect the loss of plant species to biomass, using an empirical relationship, based on a meta study of experiments, then calculate the impact of loss of biomass production on carbon storage using different scenarios. The quality is high and can be regarded for publication in Nature Communications if some assumptions and approaches are better explained or motivated.

1) The BILBI model estimate changes in species assemblages using the similarity of species composition between sites that are similar in habitat (described by land use and land cover and climate) and within the same ecoregion. Changing habitats leads to changing species composition and changing similarities. The model predicts the potential loss of species in ecoregions and sites. However, the model does not predict the potential increase of species in certain ecoregions as a result of shifting distributions of species due to climate change. An increase of species richness, which is highly possible under climate change, especially in boreal and polar ecoregions, is not included in the model. For biodiversity assessments, this is defensible as new species can be regarded as threats for original species compositions. In this case, however, it may not matter which species provide the biomass production. Can the authors modify the BILBI model to account not only for species losses but also for species gains under climate change?

2) The relationship between species richness and biomass production is derived from a meta-analysis (O'Connor et al., 2017) of experimental studies in which the number of species is manipulated. This is a very convincing study, but in this paper the relationships from O'Connor et al., is extrapolated to whole world. Generally extrapolations from experiments to the real world is not straightforward, and require testing and comparisons with data. To do such a validation is of course beyond the scope of this paper, but it should be clearly stated in the limitation that this BEF model is not validated yet. Furthermore I miss a conceptual discussion on why species richness causes higher biomass production (and is not just correlated). The mechanism is not described, but is highly needed here, as it may well be that the relationship may be the other way around in the real world. i.e.: More biomass provides more niches and hence an higher species richness. I would invite the authors to expand on this.

3) It is unclear to me how the CMIP 5 vegetation and soil carbon maps are used in the analysis. These maps are based DGVM models with changing land use and climate change. These two factors may directly affect biomass production and storage, regardless of species richness impacts. Can you better explain their role in the modelling or explain how the CMIP maps compare with the results from the BILBI model?

Some additional points:

Line 122: Figure 1 describes the modelling framework, and in the caption the authors touch upon the mechanisms they want to describe. It is however not well underpinned why a positive feedback will occur, as it may well be that LUC, CC and biodiversity loss act independently from each other on carbon storage.

Line 148 and further: The variation in the results are very high, which should a more modest set of conclusions.

Line 227 – 229: The differences estimated for the northern hemisphere between the DGVM study and this study may well be caused by not considering the potential species gains in these areas.

Reviewer #3 (Remarks to the Author):

Dear authors,

Thank you for sharing this manuscript; this was a great and inspiring read on a very important topic, in view of the coupled climate and biodiversity crisis. The paper is easy to follow (thank you for the clear explanation of the workflow!) and I also appreciate the explicit consideration of uncertainty in slope estimates. I have only a few main comments that I think would be good to address (in addition to a (long...) list of minor and editorial suggestions – hope these are helpful rather than only a pain).

Major comments

1) I'm struggling to see why species losses at the ecoregion level are connected to implications for C storage based on empirical relationships that were established at the local level. While it is nice to see the ecoregional species losses as estimated with BILBI, I cannot help but thinking that it seems much more straightforward and 'scale-consistent' to use local (grid cell) species richness loss estimates (these are also part of BILBI, no – at least to determine LU impacts?), translate these to local C storage loss, and then simply aggregate across the grid cells. Assuming there were good reasons for using the ecoregional estimates, I would very much appreciate i) a more explicit justification for the choice of regional species loss over local species loss (other than the justification provided in lines 98-100, which kind of defends the choice for regional species loss but does not make explicit why this would be preferred over local loss estimates) and ii) a comparison with C storage loss estimates based on local (grid-level) plant species loss (if feasible). I would be very interested in seeing how the estimates compare.

2) I'm not super convinced by the inclusion of soil carbon storage, given that the BEF relationship is based on above-ground biomass. Can we reasonably assume that a loss in aboveground C is directly proportional to a loss in belowground C? Perhaps this is true (cannot tell as this is not my expertise), but then at the very least there should be an explicit justification (with references) for this assumption in the paper. If this doesn't exist, then in my view it is better to just leave out the soil C part; it then seems to weaken rather than strengthen the analysis.

3) For Eq. 3 in the main text to be valid, it seems critical that the changes in above-ground biomass and the carbon stored in the vegetation (and soil) are based on the same climate and land use input data and that the carbon stored is modelled such that plant species richness plays no role in it whatsoever. I guess this is the case, but I found this a bit hard to deduce from the text. Would be good to be more explicit about this (and if these conditions are not completely fulfilled, discuss the implications thereof).

Minor comments

Line 32: biodiversity-biomass production relationships -> it's biomass stock (state) rather than production (rate), no? Same in line 114, line 199 (and perhaps more places).

Line 32-33: 'to assess the relationship between biomass (carbon storage) loss and plant biodiversity loss -

> I would say 'to assess the consequences of plant biodiversity loss for carbon storage loss'.

Line 37-39: but these estimates are not directly comparable because the scenario year differs, no? Could you get data for 2050 somehow, perhaps by using the same Cveg and Csoil projections that you now used as inputs?

Line 39: unit missing for permafrost thaw emissions

Line 63: it is a big leap from limitations of NBS initiatives to model limitations; hard to believe that the former is directly related to the latter. You may want to make a more gentle transition to the modelling here.

Line 67 -> biodiversity loss -> biodiversity.

Line 70: what relationships?

Line 70-71: why specifically over longer time frames?

Line 72: outcomes in terms of what?

Line 81-83: see my general comment re linking regional species loss to local carbon storage loss.

Line 82: change to numbered reference or at least add number? Same in line 112.

Line 94: biomass -> biomass loss.

Line 125: biodiversity -> biodiversity loss.

Line 138: would stick to one decimal.

Line 141: applied to -> combined with?

Line 152-153: but is this dependency directly proportional? See main comment above re in- or excluding soil carbon.

Line 164-170: this is very interesting! I only wonder whether it wouldn't be more consistent if you used the same projections of Cveg and Csoil to quantify carbon storage losses due to LUC and CC, and used those to put your findings in perspective? Or at least in addition to the other comparisons?

Line 198: overall -> ecoregion-level

Line 201: see earlier comments re soil carbon.

Line 206-210: but productivity (rate) and standing biomass (state) are different things, so I would avoid the suggestion of a direct comparison here.

Line 207: what relationships?

Line 212-213: which result from reference 6 are you referring to here? If I look at their Fig. 3A, their tropical hotspots of biodiv and C storage are quite well in line with your hotspots of projected CC storage loss. So is there really a contradiction here, as implied by 'however' in the next sentence?

Line 215: climate mitigation -> climate change mitigation

Line 218: what do you mean here by 'for total carbon emissions'?

Line 226: change reference Pereira 2020 to a number. Same in line 425 for Craven 2016.

Line 235-240: I perceive this para as a bit repetitive to the preceding one. Perhaps merge into one overarching para discussing both DGVMs and ESMs?

Line 256-257: I suggest using either ultimately or in the long term, not both.

Line 260: wouldn't capitalize limitations. Also, you do not quite account for these limitations in the modelling; you mainly highlight them here.

Line 271-273: this is indeed very relevant in the context of your study. Would it actually be possible to obtain an (eco)regional BEF relationship?

Line 273-274: here I get lost. Why is the time dimension relevant in this context? Could you explain?

Line 274-275: here I'm even more lost (sorry...). I don't understand how the compositional similarity modelling helps tackle the inconsistency in scale.

Line 281-283: you may add some refs to this statement. Same for line 283-284.

Line 290-293: but why this difference in time horizon? Couldn't this be harmonized? See also my earlier comments on this.

Line 293-294: again I do not understand the time dimension. Do you mean z becomes larger if we consider longer time frames? If so, how come, given that z is typically determined based on species accumulation rather than loss?

Line 300-302: this holds for SDMs; not necessarily also for your model?

Line 305: in comparisons -> in comparison

Line 320: conversion – is this the right word here? Sentence reads a bit odd.

Line 381-384: could you please provide more detail on the land use and climate change input data used for the scenarios, like data source, spatial resolution, scenario year?

Line 402: I would say that this equation represents the proportion remaining biomass ($B_{final}/B_{initial}$) rather than proportional change (which I would calculate as $\Delta B/B_{initial} = (B_{final}-B_{initial})/B_{initial}$). Would also use a different symbol than delta (which I think is most commonly understood as a subtraction).

Line 405-406: same to previous; strictly speaking your equations measure (relative) remaining biomass and (relative) remaining species richness (which are then converted to (relative) losses by taking the complement, I assume). You may want to be more precise in the wording here.

Line 414-415: I find this sentence hard to understand. What correlates with what and what bias?

Line 432-434: which scenario year? And were these projections based on the same LU and CC inputs as used for the species richness projections?

Line 441-443: this is not super convincing. First, we do not know whether the relationship between C_{soil} and plant species richness is the same as the relationship between C_{veg} and plant species richness. And second, we do not know whether the impact of LUC and CC on plant species richness translates directly to losses in soil carbon. As indicated also above, I would be inclined to leave out the soil compartment from the calculations (and touch upon the limitations/implications of omitting it in the discussion instead).

Line 462-463: ah, so here is the scenario year. Would help to mention this earlier on.

Line 464: interpretation -> interpolation?

Line 466-474: see earlier comment re equations (you need the complement of these numbers to get to losses, no?) and the uppercase delta symbol.

Figure 1: It may help a reader if you make a distinction between those links and feedbacks that are included in your modelling and those that are not. The figure seems to suggest that you account for a feedback from C storage loss to the climate (the green arrow), but you do not model this (I think), given that your scenarios are based on baseline SSP-RCP projections?

Thanks again for sharing the manuscript. Hope my comments are helpful and I look fwd to seeing this work out at some point soon.

Best wishes,
Aafke Schipper

We thank the reviewers for their very helpful comments. We have revised the paper and responded to each comment below.

REVIEWER COMMENTS

Thank you again for submitting your manuscript "Biodiversity loss reduces global terrestrial carbon storage" to Nature Communications. We have now received reports from 3 reviewers and, on the basis of their comments, we have decided to invite a revision of your work for further consideration in our journal. Your revision should address all the points raised by our reviewers (see their reports below). In particular you will need to provide further discussion justifying your approach and assumptions, exploring the limitations and how this may have affected your results. You will also need to provide context for your study and its findings as well as clarify aspects of your modelling approach. Please note that Nature Communications has a more generous word limit of 5000, space for 10 figures/tables, and allows for unlimited methods and supplementary information – so there is plenty of scope to explore all points raised to maximise the impact of your interesting study.

- Thank you for the opportunity to resubmit our manuscript, as well as the additional time to complete the revision. We have revised our manuscript to fully address the reviewer comments, as we detail below, including those related to setting the context for our paper and clarifying/justifying our modeling approach and assumptions.

Reviewer #1 (Remarks to the Author):

The subject of the manuscript entitled "Biodiversity loss reduces global terrestrial carbon storage" submitted to Nature Communications journal is very attractive and falls into the most current research issues. This study can be an inspiration for further, more detailed research.

Below I leave some suggestions/comments that may allow the authors to improve this study:

- The authors need to better identify the research gaps and clarify what are its innovative contributions to science.

- Thanks for this suggestion. We have added a research gaps table to the discussion and added a paragraph about our contributions to science to the conclusion (see below for more information).

- The authors should mention the validation process of this study.

- While it is difficult to validate the carbon estimates produced in this paper, we highlight validation performed at different steps of the analysis:
 - In the BILBI model section, we added: "GDMs perform well as predictors of species composition^{63,65}. The BILBI model relies on how species richness responds to land-use change locally, and this was tested (fit and validation) as part of the PREDICTS project⁶¹. Although more work is needed to understand the speed of extinctions following land-use change, species-area based approaches have been found to be fairly good predictors of the number of threatened or extinct species^{62,66}. Moreover,⁶⁴ found that space-for-time substitutions were 72% as accurate as time-for-time substitutions using fossil pollen records."

- Our biodiversity-biomass production relationships are based on the findings of hundreds of experiments. We have this text in our methods section: “O’Connor et al (2017) found that for primary producers, $b=0.26$ (with a 95% CI of 0.16-0.37) was valid for most assemblages and was robust to differences in experimental design and the range of species richness levels considered. While they did not find an effect of study duration on b values, previous studies have found that biodiversity-productivity relationships grow stronger over time²⁷. Although b values can vary spatially^{8,73}, there is still uncertainty in how biodiversity-ecosystem functioning relationships differ across space and in how factors like climate, environmental conditions, and species trait compositions might systematically affect the observed relationship¹³. If the places where habitat destruction is highest are also the places that tend to have the highest or lowest BEF relationships, then using a narrower range of spatially explicit values could systematically over or underestimate the carbon storage loss associated with this biodiversity loss. To address this concern, we estimated productivity losses associated with the full confidence interval range from O’Connor et al (2017). Thus, rather than considering a single slope for the BEF relationship, we consider a range of relationships that reflects variation in both composition and site-to-site differences found among previous biodiversity experiments.”
 - We have this text on our CMIP carbon model in the discussion section: “Recent analysis found that IPSL-CM5A-MR produced estimates of near-present plant carbon within the range of observation-based estimates in the non-circumpolar region, but overestimated the circumpolar regions²⁵. Thus, our carbon loss estimates from biodiversity loss may also be overestimated in these regions.”
- It is missing to indicate in Figures 2, 3, and 4 the period to which the respective losses refer.
- We have added for figure 2: “Species loss estimates are what is expected over the long term, when ecosystems approach their new equilibrium states, based on climate and land-use changes projected for 2050.” For figures 3 and 4, we added: “ Carbon loss estimates are what is expected over the long term, when ecosystems approach their new equilibrium states, based on climate and land-use changes projected for 2050.”
- The maps in Figures 1, 2, and 3 lack a graphic scale
- We added latitude and longitude to all map figures.
- In the discussion section the authors should improve the following aspects:
- (i) justify the results achieved;
- We have updated the second paragraph of the discussion to highlight that our results are similar to another study that used different biodiversity models and climate models, increasing the robustness of our findings.
- (ii) explain why these results are similar or different from previous findings; and

- We added some explanatory sentences in the discussion where we compare with other studies to highlight similarities and differences, as well as add some possible explanations.
- (iii) highlight open questions and future research directions/challenges.
- We have changed our “Accounting for limitations section” to “*Challenges and future research needs*” and added a table called “Knowledge gaps and future research directions”.

- The conclusions section should be expanded. More specifically, the authors should emphasize the contribution and implication of the study to science and recognize how this study can be useful for better land use management at the national/ regional level (and how it can be applied).

- We have expanded on the contribution to science by adding: “Our understanding of how biodiversity underpins ecosystem functions and services such as carbon storage has been increasing, but incorporating this knowledge into global projections lags behind^{29,32,61–63}. Our modeling effort provides an important example of how we can effectively link biodiversity, ecosystem functions, and ecosystem services models. As our understanding of biodiversity and ecosystem function relationships improves, our analysis can be updated to reduce the uncertainty estimates.”
- For implications for land management, we have added: “At a national and local level, this could mean that a focus on maintaining and restoring diverse ecosystems can increase the return-on-investment for carbon storage over the same land area. This may be particularly important for those ecoregions that are projected to have high levels of biodiversity-driven carbon loss.”

Reviewer #2 (Remarks to the Author):

This paper addresses a very important topic, namely the effect of biodiversity loss on ecosystem functions and services, which is the emerging field that highlight the impacts of losing biodiversity to mankind. The approach is novel as the authors link a state of art model for biodiversity with new insights of the role of biodiversity and standard models and data for carbon storage at a global level. This is not only challenging but also highly ambitious. The approach first estimates changes of plant species richness related to land use change and climate change, using the BILBI model, then connect the loss of plant species to biomass, using an empirical relationship, based on a meta study of experiments, then calculate the impact of loss of biomass production on carbon storage using different scenarios. The quality is high and can be regarded for publication in Nature Communications if some assumptions and approaches are better explained or motivated.

1) The BILBI model estimate changes in species assemblages using the similarity of species composition between sites that are similar in habitat (described by land use and land cover and climate) and within the same ecoregion. Changing habitats leads to changing species composition and changing similarities. The model predicts the potential loss of species in ecoregions and sites. However, the model does not predict the potential increase of species in certain ecoregions as a result of shifting distributions of species due to climate change. An increase of species richness, which is highly possible under climate change, especially in boreal and polar ecoregions, is not included in the model. For biodiversity assessments, this is defensible as new species can be regarded as threats for original species compositions. In this case, however, it may not matter which species provide the

biomass production. Can the authors modify the BILBI model to account not only for species losses but also for species gains under climate change?

- We realize that the way we described this in the text is a bit misleading. The BILBI model estimates species persistence at the biome level and does allow for movement between ecoregions and the possibility of some increases in species richness. While the index tends to show a loss in species persistence under most scenarios, it is in principle possible to have biodiversity gains if habitat conditions improve through time and the extent of suitable climatic conditions for a given community expands through time. However, the model does not account for the possibility of species to adapt/exploit new climatic conditions, even if it accounts for the possibility of certain locations becoming more suitable (more "similar" in GDM terms) in the future if climate becomes more similar to that currently experienced by the community. That is, because the model is based on the habitat and climate conditions where a species is currently found, the model would not predict species moving into new areas that may become suitable in the future (i.e., other habitat conditions may be suitable if experienced with the appropriate climatic conditions, but we don't know because they don't currently live in those habitats). Therefore, the model represents a somewhat pessimistic estimate of biodiversity loss from climate change. We tried to make this clearer in the discussion text.
- We have also updated the plant species loss estimates in the results section. Previously, we classified any species gains as 0% loss, but we now report the number (note that most ecoregions still experienced losses).

2) The relationship between species richness and biomass production is derived from a meta-analysis (O'Connor et al., 2017) of experimental studies in which the number of species is manipulated. This is a very convincing study, but in this paper the relationships from O'Connor et al., is extrapolated to whole world. Generally extrapolations from experiments to the real world is not straightforward, and require testing and comparisons with data. To do such a validation is of course beyond the scope of this paper, but it should be clearly stated in the limitation that this BEF model is not validated yet. Furthermore I miss a conceptual discussion on why species richness causes higher biomass production (and is not just correlated). The mechanism is not described, but is highly needed here, as it may well be that the relationship may be the other way around in the real world. i.e.: More biomass provides more niches and hence an higher species richness. I would invite the authors to expand on this.

- We have added a few sentences on the limitations of experimental data to the discussion: "Experiments can disentangle the causal effects of species richness on biomass production. However, experiments generally take place over small spatial and temporal scales and may miss important processes like dispersal, evolution, and natural patterns of species assembly and loss⁴⁶. This makes results more difficult to generalize to natural ecosystems, and additional work is needed to do so."
- We have added a few sentences on the mechanism for the BEF relationship in the introduction: "There are several possible mechanisms for this phenomenon. Species with different traits and resource requirements may be able to utilize more resources in an ecosystem through reduced competition, increased facilitation, or both, which leads to overall more efficient resource use¹⁴⁻¹⁶. At the same time, more diverse assemblages are more likely to contain the most productive species, which can increase overall functioning^{17,18}."

3) It is unclear to me how the CMIP 5 vegetation and soil carbon maps are used in the analysis. These maps are based DGVM models with changing land use and climate change. These two factors may directly affect biomass production and storage, regardless of species richness impacts. Can you better

explain their role in the modelling or explain how the CMIP maps compare with the results from the BILBI model?

- We have updated the first paragraph of step 3 in the methods section. We hope that this clarifies our use of the CMIP 5 carbon maps: "The previous step provided spatially explicit estimates of percent change in biomass associated with loss of biodiversity for each scenario. To convert biomass change to carbon storage change, we multiplied the gridded estimates of percent change in biomass from the BILBI model by a global map of terrestrial carbon stock from CMIP5. The CMIP model estimates changes in carbon storage from climate change and land use change, but does not account for changes in species richness. By using these model projections as our baseline carbon estimate, we can assess the effects of biodiversity loss on carbon storage that are expected on top of the direct changes from climate change and land use change that have already been incorporated in initial carbon storage projections."

Some additional points:

Line 122: Figure 1 describes the modelling framework, and in the caption the authors touch upon the mechanisms they want to describe. It is however not well underpinned why a positive feedback will occur, as it may well be that LUC, CC and biodiversity loss act independently from each other on carbon storage.

- Our goal for this figure was really to highlight our modeling framework, so we have tried to simplify the figure and caption by removing the discussion of positive feedback loops. We removed the arrow linking carbon emissions due to biodiversity loss to climate change so that all arrows are things that we included in our modeling process.

Line 148 and further: The variation in the results are very high, which should have a more modest set of conclusions.

- We have toned down our conclusions and highlight the large uncertainty range when we discuss potential implications of our findings.

Line 227 – 229: The differences estimated for the northern hemisphere between the DGVM study and this study may well be caused by not considering the potential species gains in these areas.

- We have added: "This could be in part because our biodiversity model may provide a more conservative estimate of potential species gains in these areas."

Reviewer #3 (Remarks to the Author):

Dear authors,

Thank you for sharing this manuscript; this was a great and inspiring read on a very important topic, in view of the coupled climate and biodiversity crisis. The paper is easy to follow (thank you for the clear explanation of the workflow!) and I also appreciate the explicit consideration of uncertainty in slope estimates. I have only a few main comments that I think would be good to address (in addition to a (long...) list of minor and editorial suggestions – hope these are helpful rather than only a pain).

Major comments

1) I'm struggling to see why species losses at the ecoregion level are connected to implications for C storage based on empirical relationships that were established at the local level. While it is nice to see

the ecoregional species losses as estimated with BILBI, I cannot help but thinking that it seems much more straightforward and 'scale-consistent' to use local (grid cell) species richness loss estimates (these are also part of BILBI, no – at least to determine LU impacts?), translate these to local C storage loss, and then simply aggregate across the grid cells. Assuming there were good reasons for using the ecoregional estimates, I would very much appreciate i) a more explicit justification for the choice of regional species loss over local species loss (other than the justification provided in lines 98-100, which kind of defends the choice for regional species loss but does not make explicit why this would be preferred over local loss estimates) and ii) a comparison with C storage loss estimates based on local (grid-level) plant species loss (if feasible). I would be very interested in seeing how the estimates compare.

- We completely agree that the local species losses are key drivers of carbon storage change in ecosystems. For this analysis, we wanted to build on the analysis in Isbell et al. (2015), which also used ecoregions as the unit of analysis for biodiversity loss. Additionally, BILBI can produce local estimates of plant species loss for land-use change scenarios, but not for climate change scenarios, as it would then not allow for movement of species with climate change. We therefore decided to keep species loss estimates at the ecoregion scale and discuss why we believe the relationship between biodiversity and productivity would hold across scales.

2) I'm not super convinced by the inclusion of soil carbon storage, given that the BEF relationship is based on above-ground biomass. Can we reasonably assume that a loss in aboveground C is directly proportional to a loss in belowground C? Perhaps this is true (cannot tell as this is not my expertise), but then at the very least there should be an explicit justification (with references) for this assumption in the paper. If this doesn't exist, then in my view it is better to just leave out the soil C part; it then seems to weaken rather than strengthen the analysis.

- We agree that the relationship between biodiversity and soil carbon is not fully established. We have now moved this finding to the discussion, noting that if biodiversity-soil carbon relationships are on a similar scale to aboveground relationships, carbon storage losses could be even higher.

3) For Eq. 3 in the main text to be valid, it seems critical that the changes in above-ground biomass and the carbon stored in the vegetation (and soil) are based on the same climate and land use input data and that the carbon stored is modelled such that plant species richness plays no role in it whatsoever. I guess this is the case, but I found this a bit hard to deduce from the text. Would be good to be more explicit about this (and if these conditions are not completely fulfilled, discuss the implications thereof).

- We tried to be as consistent as possible on the climate and land use input data used to estimate biodiversity loss and carbon storage, but model inputs were not exactly the same. We have added the following to the methods section: "We tried to be as consistent as possible on the climate and land use input data used to estimate biodiversity loss and carbon storage, but model inputs were not exactly the same. We downloaded the total carbon in vegetation (cVeg) and total carbon in soil (cSoil) layer from the CMIP5 IPSL-CM5A-MR model. The biodiversity and ecosystem services models using harmonized scenarios (BES-SIM) used climate data from either the lower resolution IPSL-CM5A-LR or WorldClim data downscaled from the IPSL-CM5A-LR depending on biodiversity model requirements. Because BILBI used 1km WorldClim data, we chose to use the mid-resolution 1.25° x 2.5° CMIP5 IPSL-CM5A-MR model to obtain higher resolution carbon maps. We obtained cVeg and cSoil for both of our scenarios – global sustainability (SSP1/RCP 2.6) and fossil-fueled development (SSP5/RCP8.5) from the Earth System Grid Federation (ESGF; <https://esgf-node.lln.gov/search/cmip5/>). The BILBI model used land use data from the Land Use Harmonization dataset version 2, while the CMIP5 IPSL-CM5A-

MR model used land use data from the Land Use Harmonization dataset version 1. Version 2 provides higher resolution data over a longer time frame with more detailed land-use categories. Although the input data is slightly different, we do not believe this invalidates the approach, as we are not comparing carbon storage changes between the models but instead using the CMIP5 IPSL-CM5A-MR model to estimate potential carbon storage under the global sustainability and fossil-fueled development scenarios, which we then use to estimate the possible magnitude of carbon storage loss driven by biodiversity loss under the same scenarios.”

Minor comments

Line 32: biodiversity-biomass production relationships -> it's biomass stock (state) rather than production (rate), no? Same in line 114, line 199 (and perhaps more places).

- Yes, in this case experiments measured biomass stock. We have changed throughout.

Line 32-33: 'to assess the relationship between biomass (carbon storage) loss and plant biodiversity loss -> I would say 'to assess the consequences of plant biodiversity loss for carbon storage loss'.

- We have updated as suggested

Line 37-39: but these estimates are not directly comparable because the scenario year differs, no? Could you get data for 2050 somehow, perhaps by using the same Cveg and Csoil projections that you now used as inputs?

- Unfortunately, we are not able to obtain estimates that are directly comparable for the same timeframe. The BILBI model uses the species-area relationship to assess plant species persistence, meaning that it projects plant species losses expected in the *long term* due to habitat conditions in a given year (e.g., poor conditions in 2050 might generate losses beyond 2050). Because the BILBI model does not predict exactly how long it will take for species to disappear once environmental conditions have changed, we do not have an exact date for the projected changes in plant persistence. This carries over into projected changes in carbon storage, since carbon storage changes as plant species are lost. This means we cannot make exact comparisons to other models that predict carbon storage changes in specific years. We now note in the text that the timescales for these projections are not exactly the same.

Line 39: unit missing for permafrost thaw emissions

- We added the unit.

Line 63: it is a big leap from limitations of NBS initiatives to model limitations; hard to believe that the former is directly related to the latter. You may want to make a more gentle transition to the modelling here.

- We changed this sentence to: “Similarly, ecosystem service models do not always account for the effects of biodiversity.”

Line 67 -> biodiversity loss -> biodiversity.

- Updated

Line 70: what relationships?

- We clarified that we are referring to biodiversity-ecosystem function relationships.

Line 70-71: why specifically over longer time frames?

- We clarified that this is because the effects of biodiversity grow stronger over time.

Line 72: outcomes in terms of what?

- We clarified that the outcomes are for marine fisheries.

Line 81-83: see my general comment re linking regional species loss to local carbon storage loss.

- As noted above, we used the regional species loss estimates to be able to incorporate climate change effects in the BILBI model. We tried to clarify this in the next paragraph as well to make this clearer.

Line 82: change to numbered reference or at least add number? Same in line 112.

- Numbers added

Line 94: biomass -> biomass loss.

- Updated

Line 125: biodiversity -> biodiversity loss.

- Updated

Line 138: would stick to one decimal.

- Updated

Line 141: applied to -> combined with?

- Updated

Line 152-153: but is this dependency directly proportional? See main comment above re in- or excluding soil carbon.

- We have removed this from the results section and now just mention it in the discussion.

Line 164-170: this is very interesting! I only wonder whether it wouldn't be more consistent if you used the same projections of Cveg and Csoil to quantify carbon storage losses due to LUC and CC, and used those to put your findings in perspective? Or at least in addition to the other comparisons?

- The BILBI model does not predict exactly how long it will take for species to disappear once environmental conditions have changed. Therefore, we do not have an exact date for the projected changes in plant persistence. This carries over into projected changes in carbon storage, since carbon storage changes as plant species are lost. This means we cannot make exact comparisons to other models that predict carbon storage changes in specific years.

Line 198: overall -> ecoregion-level

- Updated

Line 201: see earlier comments re soil carbon.

- We have removed, as noted above.

Line 206-210: but productivity (rate) and standing biomass (state) are different things, so I would avoid the suggestion of a direct comparison here.

- We rephrased to avoid the direct comparison.

Line 207: what relationships?

- We updated this sentence to: "linking species distribution models and other ecological modeling approaches with biodiversity-productivity relationships to estimate productivity changes between 2005-2070s"

Line 212-213: which result from reference 6 are you referring to here? If I look at their Fig. 3A, their tropical hotspots of biodiv and C storage are quite well in line with your hotspots of projected CC storage loss. So is there really a contradiction here, as implied by 'however' in the next sentence?

- If you look at figure 3C in reference 6, there are few areas in central Africa that came out in the 20% hotspots map, while we have quite a few in our analysis. The text also states "The Afrotropical and Indomalayan realms exhibited the lowest coverage of hotspots for both carbon and intact communities (BIp) of all realms."

Line 215: climate mitigation -> climate change mitigation

- Updated

Line 218: what do you mean here by 'for total carbon emissions'?

- We deleted this phrase.

Line 226: change reference Pereira 2020 to a number. Same in line 425 for Craven 2016.

- Updated

Line 235-240: I perceive this para as a bit repetitive to the preceding one. Perhaps merge into one overarching para discussing both DGVMs and ESMs?

- We combined these paragraphs into one.

Line 256-257: I suggest using either ultimately or in the long term, not both.

- We deleted 'ultimately'.

Line 260: wouldn't capitalize limitations. Also, you do not quite account for these limitations in the modelling; you mainly highlight them here.

- We changed this section title to "Challenges and future research needs".

Line 271-273: this is indeed very relevant in the context of your study. Would it actually be possible to obtain an (eco)regional BEF relationship?

- We agree that this would be very relevant! The data that we used here for BEF relationships comes from experiments, which generally take place over small spatial and temporal scales. Perhaps in the future modeling exercises may be able to explore regional BEF relationships, but we do not know of any that have done so at this time.

Line 273-274: here I get lost. Why is the time dimension relevant in this context? Could you explain?

- We agree that it is confusing to talk about the time dimension here and have moved this to a later paragraph where we discuss the timeframe of the projections.

Line 274-275: here I'm even more lost (sorry...). I don't understand how the compositional similarity modelling helps tackle the inconsistency in scale.

- We agree that it is confusing to talk about the compositional similarity and we have deleted it.

Line 281-283: you may add some refs to this statement. Same for line 283-284.

- References added.

Line 290-293: but why this difference in time horizon? Couldn't this be harmonized? See also my earlier comments on this.

- The BILBI model does not predict exactly how long it will take for species to disappear once environmental conditions have changed. Therefore, we do not have an exact date for the projected changes in plant persistence. This carries over into projected changes in carbon storage, since carbon storage changes as plant species are lost. This means we cannot make exact comparisons to other models that predict carbon storage changes in specific years. We tried to capture this in the text.

Line 293-294: again I do not understand the time dimension. Do you mean z becomes larger if we consider longer time frames? If so, how come, given that z is typically determined based on species accumulation rather than loss?

- What we are trying to capture here is that species-area relationships predict species loss over the long-term, but we cannot predict exactly how long it will take for species to disappear. Some species may go extinct quickly because they lose all suitable habitat, while others may go extinct over longer time periods because new conditions cannot sustain viable population sizes. Therefore, estimates produced using smaller z -values may better capture short-term changes in species, since not all species committed to extinction will have disappeared yet, while larger z -values may produce more accurate estimates in the long-term as more species go extinct. We tried to capture this in the text.

Line 300-302: this holds for SDMs; not necessarily also for your model?

- It is true that there is not a direct parallel to our model. We have modified the sentence to say: "Scenarios (including emissions and land-use) are a major source of uncertainty compared to GCMs when modeling persistence probability, and the selection of biodiversity modeling approach is also a major source of uncertainty"^{57,58}

Line 305: in comparisons -> in comparison

- Updated

Line 320: conversion – is this the right word here? Sentence reads a bit odd.

- We changed "conversion" to "towards"

Line 381-384: could you please provide more detail on the land use and climate change input data used for the scenarios, like data source, spatial resolution, scenario year?

- We added the following text: “We used land use data from the 0.25° Land Use Harmonization dataset version 2 (LUH2) ⁷⁰ and climate data from the 1km WorldClim dataset ⁷¹. For current conditions, we used the LUH2 data for 2015 and the WorldClim data for 1960-1990, and for the future conditions, we used LUH2 data for 2050 and WorldClim data for 2040-2060 ³⁵.”

Line 402: I would say that this equation represents the proportion remaining biomass ($B_{final}/B_{initial}$) rather than proportional change (which I would calculate as $\Delta B/B_{initial} = (B_{final}-B_{initial})/B_{initial}$). Would also use a different symbol than delta (which I think is most commonly understood as a subtraction).

- We updated the text to call this proportion of remaining biomass and updated the symbol to $P_{biomass}$.

Line 405-406: same to previous; strictly speaking your equations measure (relative) remaining biomass and (relative) remaining species richness (which are then converted to (relative) losses by taking the complement, I assume). You may want to be more precise in the wording here.

- We updated the text to say that we are assessing the proportion of remaining biomass from the proportion of remaining plant species richness.

Line 414-415: I find this sentence hard to understand. What correlates with what and what bias?

- We have reworded this sentence to: “If the places where habitat destruction is highest are also the places that tend to have the highest or lowest BEF relationships, then using a narrower range of spatially explicit values could systematically over or underestimate the carbon storage loss associated with this biodiversity loss”.

Line 432-434: which scenario year? And were these projections based on the same LU and CC inputs as used for the species richness projections?

- We moved the sentence about the scenario year to this paragraph and clarified the differences in projections for the LU and CC inputs in the species richness and carbon storage projections.

Line 441-443: this is not super convincing. First, we do not know whether the relationship between Csoil and plant species richness is the same as the relationship between Cveg and plant species richness. And second, we do not know whether the impact of LUC and CC on plant species richness translates directly to losses in soil carbon. As indicated also above, I would be inclined to leave out the soil compartment from the calculations (and touch upon the limitations/implications of omitting it in the discussion instead).

- We have reframed this section to say that we assess how large soil carbon losses could be if plant biodiversity-soil carbon relationships are on a similar magnitude to aboveground biomass. We then just present these results in the discussion/supplemental figures rather than as a main finding in the results.

Line 462-463: ah, so here is the scenario year. Would help to mention this earlier on.

- We moved this sentence earlier.

Line 464: interpretation -> interpolation?

- Corrected

Line 466-474: see earlier comment re equations (you need the complement of these numbers to get to losses, no?) and the uppercase delta symbol.

- Thanks for catching this! Yes, it is the complement and we have updated the equations accordingly. In this case, we believe the delta symbol makes sense because it is a subtraction.

Figure 1: It may help a reader if you make a distinction between those links and feedbacks that are included in your modelling and those that are not. The figure seems to suggest that you account for a feedback from C storage loss to the climate (the green arrow), but you do not model this (I think), given that your scenarios are based on baseline SSP-RCP projections?

- Since the goal of this figure was to show our modeling framework, we deleted the part of the arrow linking carbon emissions due to biodiversity loss to climate change.

Thanks again for sharing the manuscript. Hope my comments are helpful and I look fwd to seeing this work out at some point soon.

Best wishes,
Aafke Schipper

REVIEWERS' COMMENTS

Reviewer #2 (Remarks to the Author):

All remarks and concerns are well addressed in the revision, only one remark remain. On page 2 lines 54-65 the authors describe the possible mechanism why higher biodiversity causes higher rate of carbon sequestration. Although this is plausible, but mainly based on correlation. The other explanation is that more biomass will lead to higher biodiversity. Is there any evidence that rules out that potential mechanism for the observed relationship?

Another short remark is on page 6 lines 155 and 156, where numbers are given for the parameters b and z , without reference to a corresponding formula. I might have missed it, but it would be good to make a clear reference to the formulas used.

Reviewer comments

- On page 2 lines 54-65 the authors describe the possible mechanism why higher biodiversity causes higher rate of carbon sequestration. Although this is plausible, but mainly based on correlation. The other explanation is that more biomass will lead to higher biodiversity. Is there any evidence that rules out that potential mechanism for the observed relationship?
 - While it is true that in natural systems, most evidence between biodiversity and biomass is correlative, the advantage of experiments is that species richness is directly manipulated. Biodiversity experiments have rigorously quantified the effects of richness and composition by randomizing both how many and which species are included in each experimental plot (Tilman et al. 2014 - <https://www-annualreviews-org.usgslibrary.idm.oclc.org/doi/10.1146/annurev-ecolsys-120213-091917>). This creates a gradient of plant species richness without systematically changing which species are present at each level of plant diversity. Therefore, variation in ecosystem function between experimental levels of richness indicates an effect of plant diversity that is independent of any effects of changing which species are present. Variation in ecosystem function within an experimental level of richness indicates an effect of species composition that is independent of any effects of species richness. We have a sentence about this lower in the introduction: “The advantage of using the local experimental data is that by strictly controlling for species richness, composition, and other confounding factors, local experiments can disentangle the causal effects of species richness on biomass production.”
- Another short remark is on page 6 lines 155 and 156, where numbers are given for the parameters b and z , without reference to a corresponding formula. I might have missed it, but it would be good to make a clear reference to the formulas used.
 - Thank you for pointing this out! We added a brief description of b and z in the results when they first appear with a note to see methods section for more information on these variables.